# BroRL: Scaling Reinforcement Learning via Broadened Exploration

## Abstract

Reinforcement Learning with Verifiable Rewards (RLVR) has emerged as a key ingredient for unlocking complex reasoning capabilities in large language models. Recent work ProRL (Liu et al., 2025a) has shown promise in scaling RL by increasing the number of training steps. However, performance plateaus after thousands of steps, with clear diminishing returns from allocating more computation to additional training. In this work, we investigate a complementary paradigm for scaling RL: **BroRL**—increasing the number of rollouts per example to hundreds to exhaustively **Bro**aden exploration, which yields continuous performance gains beyond the saturation point observed in ProRL when scaling the number of training steps. Our approach is motivated by a mass balance equation analysis allowing us to characterize the rate of change in probability mass for correct and incorrect tokens during the reinforcement process. We show that under a one-step RL assumption, sampled rollout tokens always contribute to correct-mass expansion, while unsampled tokens outside rollouts may lead to gains or losses depending on their distribution and the net reward balance. Importantly, as the number of rollouts per example $N$ increases, the effect of unsampled terms diminishes, ensuring overall correct-mass expansion. To validate our theoretical analysis, we conduct simulations under more relaxed conditions and find that a sufficiently large rollout size $N$—corresponding to ample exploration—guarantees an increase in the probability mass of all correct tokens. Empirically, BroRL revives models saturated after 3K ProRL training steps and demonstrates robust, continuous improvement, achieving state-of-the-art results for the 1.5B model across diverse benchmarks. Notably, under the same training time, BroRL is both more data- and compute-efficient: large-$N$ rollouts reduce the number of filtered samples during dynamic sampling at the algorithmic level and shift generation from memory-bound to compute-bound at the hardware level, nearly doubling throughput compared to ProRL in our hardware setup, highlighting BroRL's practicality for real-world deployment.

## 1 Introduction

One of the central drivers behind the rapid advances in Large Language Models (LLMs) over the past a few years has been the discovery and application of *Scaling Laws*. Kaplan et al. (2020) showed that model performance follows predictable power-law improvements with respect to parameters, data, and compute. Building on this, Hoffmann et al. (2022) demonstrated that training is compute-optimal when model size and training tokens are scaled proportionally. These insights powered breakthroughs from GPT-3 to Claude/GPT-4 era, where scaling laws guided compute-optimal training of larger and more capable models.

More recently, Reinforcement Learning with Verifiable Rewards (RLVR) has brought new excitement to the field, unlocking complex reasoning in LLMs and fueling the rise of large reasoning models such as DeepSeek-R1 (Guo et al., 2025) and OpenAI-o3 (Jaech et al., 2024). Yet, *how to effectively scale the RLVR paradigm* remains an open question. Recent work ProRL (Liu et al., 2025a; Hu et al., 2025b) has demonstrated the potential of scaling RL by increasing the number of training steps. While this approach yields steady initial gains, performance plateaus after thousands of steps, with clear diminishing returns from allocating more computation to additional training.

In this work, we investigate a complementary dimension of the RL scaling law: **BroRL**—increasing the number of rollouts per example to the order of hundreds or thousands to exhaustively **Bro**aden exploration. Intuitively, our approach mirrors how humans tackle hard problems (e.g., four color theorem), making countless attempts over decades until a breakthrough emerges. Theoretically, our approach is motivated by a mass balance equation analysis. As shown in Figure 2, under the one-step RL assumption, the change in correct-token probability mass $\Delta Q_{\text{pos}}$ consists of two parts. (1) The sampled portion always contributes a non-negative gain by promoting sampled-correct tokens and demoting sampled-incorrect tokens, thus ensuring mass expansion. (2) The unsampled portion is conditional, potentially adding or removing mass depending on the batch distribution. Importantly, as the number of rollouts per prompt $N$ increases, the influence of the unsampled terms diminishes, driving the overall effect toward $\Delta Q_{\text{pos}} \geq 0$.

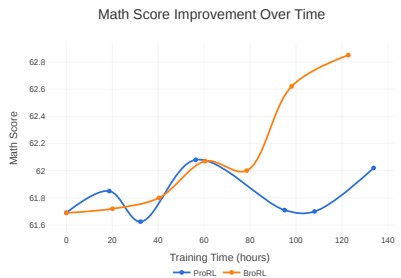

Figure 1: Empirical results demonstrate that BroRL ($N = 512$) continues to improve math performance, whereas ProRL ($N = 16$) reaches a plateau at the 3k-steps checkpoint and further degrades with prolonged training.

To verify our theoretical analysis, we conduct simulations with a TRPO-style linear surrogate objective. The results show that a sufficiently large rollout size $N$—corresponding to ample exploration—guarantees an increase in the probability mass of all correct tokens and eliminates knowledge shrinkage (Wu et al., 2025), i.e., worst-case probability reductions among correct tokens, which implies that with enough exploration RLVR can reliably acquire new knowledge without forgetting the old. Building on this foundation, we apply the BroRL recipe to scale RL training on real-world reasoning models. In particular, we continue training the ProRL model that plateaues after 3K steps and find that BroRL yields robust, continuous performance improvements, ultimately achieving new state-of-the-art results for the 1.5B model across diverse benchmarks.

Notably, under the same training time, BroRL is both more data- and compute-efficient: large-$N$ rollouts reduce the number of filtered samples during dynamic sampling at the algorithmic level and shift generation from memory-bound to compute-bound at the hardware level, nearly doubling throughput compared to ProRL in our hardware setup, underscoring BroRL's practicality for real-world deployment. BroRL highlights the central role of exploration in RL, revealing that the per-

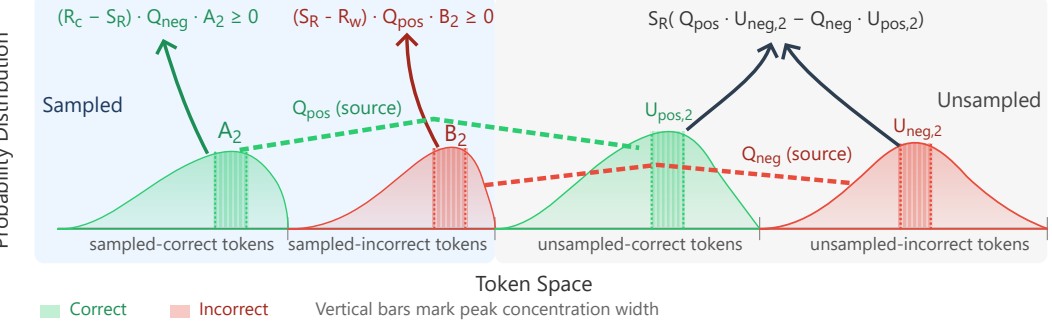

Figure 2: This illustration shows how a single RLVR update step alters the total probability mass $\Delta Q_{\text{pos}}$ for correct tokens, where the dashed guide lines labeled $Q_{\text{pos}}$ (green) and $Q_{\text{neg}}$ (red) connect the pooled probability assigned to the correct and incorrect token sets across sampled and unsampled regions. The change is composed of two parts: the Sampled portion (left) always produces a non-negative gain by promoting "sampled-correct" tokens (concentration measured by $A_2$) and demoting "sampled-incorrect" tokens (concentration measured by $B_2$), thereby shifting probability from the $Q_{\text{neg}}$ pool to the $Q_{\text{pos}}$ pool. The unsampled part (right) is conditional: it can add or remove mass depending on the batch mood $S_R$ and whether unsampled incorrect probability is more concentrated than unsampled correct probability. As the number of samples per prompt $N$ grows, the unsampled concentration terms $U_{\text{pos},2}$ and $U_{\text{neg},2}$ shrink, so the net effect tends toward $\Delta Q_{\text{pos}} \geq 0$; the amount of mass moved scales with the pool sizes $Q_{\text{pos}}$ and $Q_{\text{neg}}$.

ceived limits of RLVR are sometimes artifacts of algorithmic design (e.g., insufficient rollouts) rather than the fundamental limits of RL itself, underscoring the necessity and promise of future algorithmic advances in RL.

## 2 THEORETICAL ANALYSIS

We develop a theoretical analysis based on a mass balance argument, common in physics for mass and transfer analysis. Our analysis is performed in the logit domain, focusing on the partial mass of correct tokens (negative tokens, respectively). By a common abuse of language, we shall regularly use "probability" to refer to a logit [1].

**Notation.** We consider a vocabulary of size $V$, with logits $z \in \mathbb{R}^V$ and probabilities $p = \text{softmax}(z)$. Let $\mathcal{P}$ and $\mathcal{N}$ denote the sets of correct and incorrect tokens in vocabulary $V$, respectively. $N$ rollout tokens is sampled, where each sampled token receives a binary reward $R_i \in \{R_c, R_w\}$ depending on whether it is correct or incorrect, while unsampled tokens are assigned $R_i = 0$. In the standard setting, the rewards satisfy $R_c \geq 0 \geq R_w$. Let $A \subseteq \mathcal{P}$ be the set of sampled correct tokens, $B \subseteq \mathcal{N}$ the set of sampled incorrect tokens, and $U$ the set of unsampled tokens.

Let the partial mass $P_{\text{pos}}$ denote the total probability mass of the sampled correct tokens, and $P_{\text{neg}}$ the total probability mass of the sampled incorrect tokens. Similarly, let $Q_{\text{pos}}$ be the total probability mass of all correct tokens, and $Q_{\text{neg}}$ the total probability mass of all incorrect tokens.

$$P_{\text{pos}} = \sum_{i \in A} p_i, \quad P_{\text{neg}} = \sum_{i \in B} p_i, \quad Q_{\text{pos}} = \sum_{i \in \mathcal{P}} p_i, \quad Q_{\text{neg}} = 1 - Q_{\text{pos}}.$$

The corresponding second moments, which measure how each partial mass is concentrated, are given by:

$$A_2 = \sum_{i \in A} p_i^2, \quad B_2 = \sum_{i \in B} p_i^2, \quad U_{\text{pos},2} = \sum_{i \in U \cap \mathcal{P}} p_i^2, \quad U_{\text{neg},2} = \sum_{i \in U \cap \mathcal{N}} p_i^2.$$

Finally, define $S_R = \sum_{k \in \mathcal{A}} R_c \, p_k + \sum_{k \in \mathcal{B}} R_w \, p_k = R_c \, P_{\text{pos}} + R_w \, P_{\text{neg}}$ which represents the net contribution of sampled tokens, balancing the rewards from correct and incorrect tokens.

**Connection between** $\text{pass}@k$ **and** $Q_{\text{pos}}$   The quantity $Q_{\text{pos}}$ denotes the total probability mass assigned to correct tokens. For a single task, let $Q_{\text{pos}}(x) \in [0, 1]$ denote the total probability mass assigned to correct tokens for input $x$. When drawing $k$ i.i.d. samples, the per-task success probability for input $x$ is

$$\text{pass}@k(x) = 1 - \left(1 - Q_{\text{pos}}(x)\right)^k.$$

This expression is strictly increasing in $Q_{\text{pos}}(x)$; thus, RLVR updates that increase the positive probability mass directly improve $\text{pass}@k$, and at a geometric rate. Taking the expectation over the task distribution yields

$$\mathbb{E}_x[\text{pass}@k(x)] = 1 - \mathbb{E}_x\left[(1 - Q_{\text{pos}}(x))^k\right].$$

Moreover, if $Q_{\text{pos}}(x)$ increases pointwise (i.e., $Q'_{\text{pos}}(x) \geq Q_{\text{pos}}(x)$ for all $x$, with strict inequality on a set of positive measure), then both $\text{pass}@k(x)$ and its expectation increase strictly.

**One-step RLVR update.**   We perform our analysis under the simplifying assumption of a single step of RLVR, which allows us to obtain convenient analytical formulas. We model a single RLVR step as adjusting logits $z \in \mathbb{R}^V$ via a gradient update with rewards $\{R_c, R_w\}$ on sampled tokens. The update induces a first-order change in token probabilities $\Delta p$, which aggregates into a total correct-mass change

$$\Delta Q_{\text{pos}} = \sum_{i \in \mathcal{P}} \Delta p_i,$$

where $\mathcal{P}$ is the set of correct tokens. Then we show that the one-step change decomposes into an *sampled positive* term (always nonnegative) and an *unsampled coupling* term (which can be negative

---

[1]This language is unrelated to the confidence we may assign to a logit and whether the model is statistically calibrated or not (Geng et al., 2023; Liu et al., 2025b).

but vanishes as the rollout size $N$ grows). This decomposition allows us to uncover *distinct dynamics* in each term. In particular, the scaling of each of these terms with respect to $N$, leads us to identify the *roll-out size* as a key quantity to strike a trade-off for superior performance in experiments.

Formally:

**Theorem 1 (Sign of Correct-Mass Change).**

$$\Delta Q_{\text{pos}} = \frac{\eta}{N}\Big[(R_c - S_R)Q_{\text{neg}}A_2 \; + \; (S_R - R_w)Q_{\text{pos}}B_2 \; + \; S_R\big(Q_{\text{pos}}U_{\text{neg},2} - Q_{\text{neg}}U_{\text{pos},2}\big)\Big],$$

*where $A_2, B_2 \geq 0$, and $S_R \in [R_w, R_c]$, which implies $R_c - S_R \geq 0$ and $S_R - R_w \geq 0$. Therefore, the first two terms nonnegative, While the last term represents the coupling of unsampled masses.*

**Interpretation.** Three terms account for the change in the probability mass of correct predictions, as illustrated in Figure 2.

The first term, $(R_c - S_R)Q_{\text{neg}}A_2$, arises from sampled-correct tokens. Each correct token has an advantage of $(R_c - S_R)$, meaning it is explicitly rewarded. Normalization redistributes this reward by taking mass from the incorrect pool (the $Q_{\text{neg}}$ share), and the size of the effect grows when those correct tokens are highly concentrated (large $A_2$). This term is always nonnegative: pushing up correct tokens can never reduce correct probability. This is a key feature of the sampled-correct tokens component of the reinforcement dynamics.

The second term, $(S_R - R_w)Q_{\text{pos}}B_2$, arises from sampled-incorrect tokens. These have an effective (negative) advantage of $(R_w - S_R) \leq 0$, so their probabilities are pushed down. Normalization then routes the freed-up mass to the correct pool in proportion to its size ($Q_{\text{pos}}$), and the effect is stronger when the incorrect samples were concentrated (large $B_2$). Again, this is always nonnegative: pushing down incorrect tokens leaves more probability for correct ones.

The third term, $S_R\big(Q_{\text{pos}}U_{\text{neg},2} - Q_{\text{neg}}U_{\text{pos},2}\big)$, comes from unsampled tokens, and unlike the first two, it can be positive or negative. Here the batch "mood" $S_R$ sets the direction: If $S_R > 0$ (a reward-positive batch), unsampled logits are nudged downward. This helps if unsampled incorrect mass is more concentrated (large $U_{\text{neg},2}$), but hurts if unsampled correct mass is more concentrated (large $U_{\text{pos},2}$). If $S_R < 0$ (a reward-negative batch), the signs flip: unsampled logits are nudged upward. This helps if unsampled correct tokens are more concentrated, but hurts if unsampled incorrect mass dominates.

Thus, the first two terms always contribute positively, while the third can either reinforce or oppose them depending on batch balance and how probability is distributed among unsampled tokens.

We draw several implications: (i) As the per-prompt sampling size $N$ grows, the unsampled terms $U_{\text{pos},2}, U_{\text{neg},2}$ shrink, ensuring $\Delta Q_{\text{pos}} \geq 0$. (ii) Even for small $N$, positivity holds under balanced batches ($S_R \approx 0$) or when unsampled mass is sufficiently small. (iii) Increasing per-prompt sampling size $N$ directly improves pass@k by enlarging the positive margin of $\Delta Q_{\text{pos}}$.

Since pass@k$(x)$ is monotone in $Q_{\text{pos}}(x)$, any step with $\Delta Q_{\text{pos}} > 0$ improves success probability. Larger $N$ strengthens this effect by reducing the contribution of the third (unsampled) term, which can be negative under certain conditions. Full derivations and proofs are in Appendix D.1.

**Expected decay of unsampled mass.** The coupling term in Theorem 1 depends on the unsampled second moments $U_{\text{pos},2}, U_{\text{neg},2}$. These shrink as the rollout size $N$ grows:

**Lemma 2.** *Let a token with probability $p$ be sampled independently in each of $N$ draws. The expected "unsampled second-moment" contribution of this token is*

$$\mathbb{E}[U_2(p)] \; = \; p^2(1-p)^N.$$

*Corollary 3.* For a collection of tokens with probabilities $\{p_i\}$, the expected total unsampled second moment after $N$ draws is
$$\sum_i p_i^2(1-p_i)^N.$$

By linearity, this ensures $U_{\text{pos},2}$ and $U_{\text{neg},2}$ decrease monotonically with $N$, driving $\Delta Q_{\text{pos}}$ toward nonnegativity as $N$ increases. A full proof is provided in Appendix D.2.

## 3 BroRL: Broad Reinforcement Learning

### 3.1 Background: Prolonged Reinforcement Learning

We adopt the prolonged reinforcement learning (RL) framework from ProRLv2 (Hu et al., 2025b). This approach is centered around a clipped Proximal Policy Optimization (PPO) algorithm, with the objective function:

$$\mathcal{L}_{\text{PPO}}(\theta) = \mathbb{E}_\tau \left[ \min \left( r_\theta(\tau) A(\tau), \text{clip}(r_\theta(\tau), 1 - \varepsilon_{\text{low}}, 1 + \varepsilon_{\text{high}}) A(\tau) \right) \right],$$

where $r_\theta(\tau)$ is the probability ratio and $A(\tau)$ is the advantage. A key feature is its REINFORCE++-style decoupled advantage normalization (Hu et al., 2025a). First, the advantage $A_\tau$ for a trajectory $\tau$ with return $R_\tau$ is computed by subtracting the mean return of its corresponding group for each prompt. This value is then normalized across the entire global sample batch:

$$A_\tau = R_\tau - \text{mean}_{\text{group}}(R_\tau),$$

$$A_\tau^{\text{norm}} = \frac{A_\tau - \text{mean}_{\text{batch}}(A_\tau)}{\text{std}_{\text{batch}}(A_\tau)}.$$

To further improve performance and exploration, the framework integrates several key techniques. A core component is *Dynamic Sampling* (Yu et al., 2025), which filters out trivial trajectories that are either entirely correct or entirely incorrect to focus training on the most informative samples. For a batch $\mathcal{B}$ of trajectories $\tau$, the filtered batch $\mathcal{B}'$ is:

$$\mathcal{B}' = \left\{ \tau \in \mathcal{B} \; \middle| \; 0 < \sum_{i=1}^{N} \mathbb{I}(M_i = M_{\text{correct}}) < N \right\},$$

where $N$ is the number of rollout samples per prompt, $M_i$ is the prediction and $\mathbb{I}(\cdot)$ is the indicator function. Other methods include periodic resets of the reference policy, exploration enhancements via Clip-Higher ($\varepsilon_{\text{high}} > \varepsilon_{\text{low}}$) (Yu et al., 2025), and truncated importance sampling (Yao et al., 2025) to correct off-policy mismatch between the inference engine and the training engine.

### 3.2 Scaling Reinforcement Learning via Number of Rollouts

BroRL is predicated on the principled scaling of the rollout size per prompt $N$, which directly operationalizes the theoretical insights established in Section 2. The decomposition in Theorem 1 reveals that the policy update, as measured by the change in correct probability mass ($\Delta Q_{pos}$), is subject to a potentially negative "unsampled coupling" term, $S_R(Q_{\text{pos}}U_{\text{neg},2} - Q_{\text{neg}}U_{\text{pos},2})$, which can introduce instability and counteract policy improvement. Our theoretical framework rigorously establishes that the detrimental influence of this term on $\Delta Q_{\text{pos}}$ is inversely related to the rollout size $N$. Consequently, in contrast to conventional approaches, BroRL employs a significantly large $N$ to substantially increase the rollout diversity for each prompt. This rollout size $N$ scaling robustifies the learning signal by minimizing the variance and potential negativity arising from unsampled portions of the action space. This ensures a more consistent and stable policy optimization process, directly translating our theoretical guarantees into a more effective training regime for complex reasoning tasks.

## 4 Experiments

We first conduct token-level simulations to verify our theoretical insight (Section 4.1), and then apply BroRL in real-world scenarios by continuing RL training on ProRL models that plateau after 3K steps (Section 4.2).

### 4.1 Simulation of the Theoretical Analysis

**Simulation Setup.** We build a token-level simulator reflecting the per-token update analysis in Theorem 1, using a TRPO-style linear surrogate objective (Schulman et al., 2015; Zhu et al., 2025). The vocabulary has size $d = 128{,}000$, with a subset $\mathcal{P} \subset [d]$ of 10,000 correct tokens assigned reward $R_i = +1$ and the remainder $R_i = -1$.

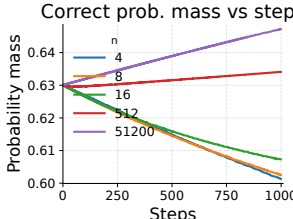 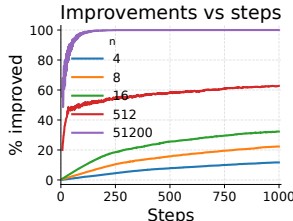 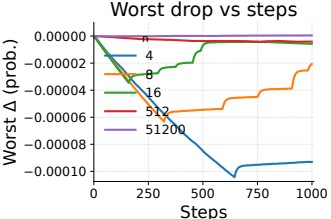

Figure 3: Training dynamics of the simulator under varying rollout size $N$. We track (i) the total probability mass assigned to correct actions, (ii) the fraction of correct actions whose probability increased relative to step 0, and (iii) the worst negative change in probability among correct actions. Larger $N$ produces more stable updates, faster accumulation of probability mass, and crucially it eliminates knowledge shrinkage by removing negative probability drops altogether.

Logits $z \in \mathbb{R}^d$ are initialized as $z_i = 0$, with optional seeding by setting $z_i = 3$ for $i \in \mathcal{P}$ and fixing one anchor token $z_0 = 5$. Probabilities are $p_i = \text{softmax}(z/\tau)_i$ with $\tau = 1$. At each step $t$, we draw $N$ i.i.d. samples center rewards with the batch baseline $b = \frac{1}{N}\sum_{j=1}^{N} R_j$, yielding $\tilde{r}_j = R_j - b$ to reduce variance.

We optimize the RLVR surrogate

$$\mathcal{L}_{\text{sur}} = -\frac{1}{N}\sum_{j=1}^{N} \tilde{r}_j \, p_j$$

and update $z$ with AdamW (learning rate $\eta = 10^{-3}$) for $T = 1000$ steps. Rollout size $N \in \{4, 8, 16, 512, 51200\}$ are varied while all other hyperparameters are fixed.

After each update, we record: (i) the *total correct probability mass* $Q_{\text{pos}} = \sum_{i \in \mathcal{P}} p_i$, (ii) the *percent of correct tokens* whose probabilities increased relative to step 0, and (iii) the *worst probability drop* among correct tokens.

**Results.** The simulation results align with our key insight: increasing rollout size $N$ dampens the influence of the unsampled coupling term in Theorem 1, yielding more reliably positive mass expansion and stable policy updates. As shown in Figure 3, larger rollout size $N$ accelerates the growth of positive mass $Q_{\text{pos}}$ and increase the proportion of correct tokens whose probabilities improve at each step, whereas small-$N$ updates exhibit slower progress, higher variance, and occasional regressions.

Importantly, the worst-case probability drops among correct tokens—known as knowledge shrinkage (Wu et al., 2025) and common with small $N$— disappear at large $N$. In the extreme, when $N$ is very large, RLVR eliminates knowledge shrinkage entirely, ensuring that all correct tokens gain probability mass. This matches the theoretical prediction that unsampled second-moment terms shrink with width (Lemma 2), thereby suppressing potential harmful contributions from unsampled tokens. Taken together, these findings confirm that allocating compute to rollout size, rather than step depth, yields consistently positive updates and provides the principled basis for BroRL.

## 4.2 EMPIRICAL STUDY ON LARGE LANGUAGE MODELS

### 4.2.1 EXPERIMENTAL SETUP

**Base Model.** We build upon the publicly available ProRLv2 checkpoint and five task families: math, code, science, IFEval (Zhou et al., 2023) and reasoning gym (Stojanovski et al., 2025). This model, having already undergone 3,000 RL training steps with a context length of 8,192 tokens, provides a strong starting point. To further enhance its capabilities, especially for tasks requiring long-context reasoning, we expanded its context window to 16,384 tokens for all subsequent training phases with 64 NVIDIA H100 GPUs and the veRL famework (Sheng et al., 2025).

**BroRL Implementation.** We continue RL training on top of ProRLv2 checkpoint with the BroRL recipe. We increased the number of generated samples per prompt from the baseline of 16 to $N = 512$. This large value of $N$ is central to our hypothesis that a broader exploration of the solution space

during each update step leads to more robust and generalizable reasoning abilities. For baseline comparison, we also extend RL training on top of ProRLv2 checkpoint using the original ProRL recipe under the same compute budget.

**Learning Rate Scaling.** To maintain training stability while accommodating the significantly larger effective batch size resulting from the increased rollout size ($N$), we adjusted the learning rate while keeping the number of PPO mini-batches per step unchanged. Specifically, the learning rate was scaled proportionally to the square root of the increase in the batch size (Krizhevsky, 2014). Let $\eta_0$ be the base learning rate for a reference batch size $B_0$. Our new learning rate $\eta_{\text{new}}$ for a new, larger batch size $B_{\text{new}}$ is determined by the formula: $\eta_{\text{new}} = \eta_0 \times \sqrt{\frac{B_{\text{new}}}{B_0}}$ . This principled adjustment ensures that the magnitude of parameter updates remains well-controlled.

### 4.2.2 ANALYSIS OF PASS@1 SUCCESS RATE

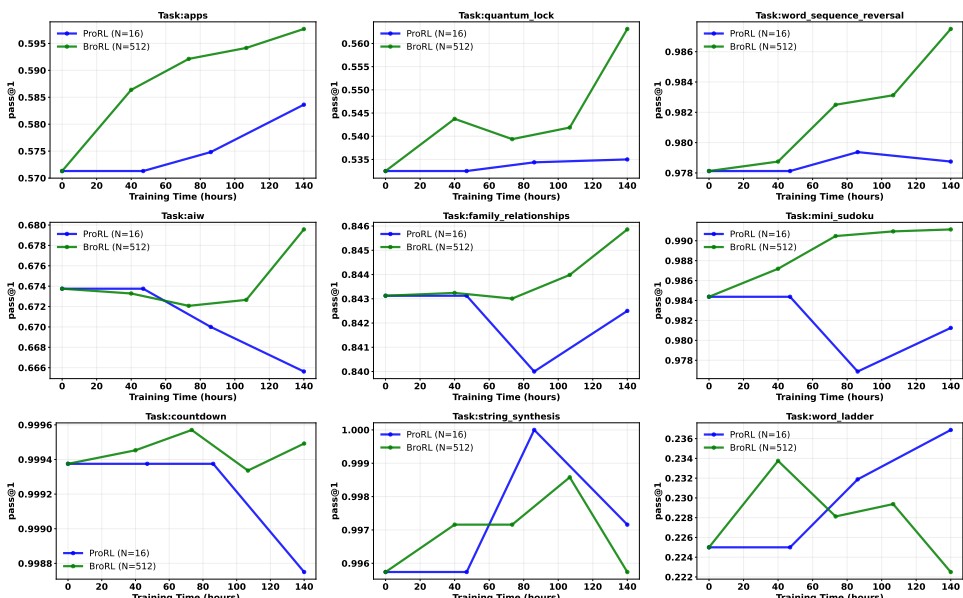

Figure 4: Pass@1 comparison of BroRL vs. ProRL, normalized by training compute. Rows show representative trajectories: (1) both improve but BroRL consistently outperforms ProRL; (2) ProRL degrades while BroRL continues to improve; (3) both methods fail to yield consistent gains.

To better understand the practical impact of our methods, we compare BroRL and ProRL across benchmark tasks under equalized training compute. Figure 4 summarizes these results by tracking performance at intermediate checkpoints. We observe three characteristic types of training trajectories. In the first, both methods improve but BroRL consistently outperforms ProRL, aligning with theoretical expectations and highlighting stronger learning dynamics. In the second, ProRL degrades over time while BroRL continues to improve, underscoring its robustness. In the third, both methods fail to achieve consistent gains, In the third scenario, both methods fail to achieve consistent gains, suggesting that $N = 512$ might not be large enough for some of the harder problems. Most benchmarks fall into the first two patterns, while the third is less common. Collectively, these trajectories show that BroRL not only matches theoretical predictions but also demonstrates clear practical advantages in training efficiency and stability. Importantly, all results are measured on the test dataset, highlighting that BroRL's improvements reflect not only better learning dynamics during training but also stronger generalization to unseen instances.

To complement the trajectory analysis, we perform a statistical evaluation to test whether BroRL provides a measurable improvement over ProRL. We collect results from all individual problem instances across benchmarks, yielding over $10,000$ data points, and measure Pass@1 at the final checkpoint under equal training compute ($\sim 140$ hours). A paired t-test reveals a small but statistically significant advantage for BroRL ($\Delta = 0.0033$, $t = 4.84$, one-tailed $p = 6.5 \times 10^{-7}$).

One-tailed and t-tests reject the null hypothesis, confirming that BroRL outperforms ProRL with extremely strong statistical confidence. Although the mean difference is small, this is expected since we build on a strong baseline already fine-tuned for 3000 steps and evaluate after only 100 additional steps, where gains scale roughly log-linearly with training time (Liu et al., 2025a). In this regime, even a modest but statistically significant improvement is meaningful, confirming that BroRL yields more reliable progress and better generalization to unseen test instances.

### 4.2.3 Pushing Reasoning Boundaries Beyond Steps Scaling

A common challenge in longterm RLVR training is performance saturation, where simply training longer steps yields diminishing returns. The initial ProRLv2 checkpoint trained 3000 RL steps had reached such a plateau. This section details a controlled experiment to demonstrate that BroRL's rollout-scaling approach is not only more effective but also more time-efficient at breaking through this performance ceiling.

We compare the performance of two continued training strategies. The ProRL approach uses a conventional small rollout size ($N = 16$), while our BroRL approach scales the size significantly ($N = 512$). Table 1 and Figure 1 summarize the trade-offs in terms of computational cost and performance outcome at different checkpoints.

Table 1: Efficiency and Performance Comparison. BroRL shows steady improvement and achieves a higher score in less total time, while the ProRL stagnates and degrades. The number of samples refers to the amount before dynamic sampling filtering.

| Method | N | Prompts / Step | RL Steps | Samples (k) | Time (h) | Math Score | Code Score | Reasoning Gym Score |
|---|---|---|---|---|---|---|---|---|
| Baseline | 16 | 512 | 3000 | - | - | 61.69 | 52.00 | 61.29 |
| ProRL | 16 | 512 | +225 | +4390 | 56.3 | 62.08 | 52.26 | 62.10 |
| ProRL | 16 | 512 | +535 | +10439 | 133.8 | 62.02 | 52.74 | 61.45 |
| BroRL | 512 | 128 | +107 | +11226 | 98.1 | 62.62 | 53.31 | 62.71 |
| BroRL | 512 | 128 | +134 | +14059 | 122.8 | 62.85 | 53.48 | 62.82 |
| BroRL | 512 | 128 | +191 | +20039 | 173.8 | **63.03** | **54.20** | **63.09** |

The result reveals two divergent outcomes. The ProRL method shows marginal initial gains across all tasks, peaking at 62.08 on Math and 62.10 on Reasoning Gym. However, continued training leads to performance stagnation and degradation. While the Code Score sees a minor increase to 52.74, the Math Score drops to 62.02, and the Reasoning Gym Score falls significantly to 61.45. This pattern, observed after nearly 134 hours, clearly illustrates the diminishing and ultimately negative returns of simply scaling training steps for this saturated model.

In stark contrast, the BroRL approach demonstrates robust and continuous improvement across all three benchmarks, ultimately achieving the highest scores: 63.03 in Math, 54.20 in Code, and 63.9 in Reasoning Gym. The efficiency of this rollout-size-scaling approach is particularly striking. After just 98.1 hours, BroRL had already decisively surpassed the final performance of the ProRL method across all metrics, doing so in approximately 35 fewer hours. This result confirms that scaling the rollout size $N$ is a more effective and computationally efficient strategy for pushing the performance boundaries of a saturated model. This superior performance stem not from performing more gradient updates, but from executing fewer, yet higher-quality updates, as we maintain the same number of PPO mini-batches per RL step. More evaluation details and results are in Appendix E. The following section investigates the core reasons for this enhanced efficiency at both the algorithmic and hardware levels.

### 4.2.4 Rollout size scaling's Impact on GPU Compute Efficiency

The primary performance bottleneck in training models for long Chain-of-Thought (CoT) reasoning via RLVR is the sample generation phase (Hu et al., 2024). Our BroRL framework address this challenge through a two-pronged approach: one at the algorithmic level and another at the hardware level. To isolate these variables, all experiments were conducted on an identical hardware setup (GPU and node count). Table 2 quantifies these two factors.

First, at the algorithmic level, a larger rollout size $N$ leads to a more diverse set of candidate samples. The *Dynamic Sampling Pass Rate* in Table 2 shows that with $N = 512$, 62% of the generated

Table 2: Algorithmic and Hardware Efficiency Metrics. BroRL improves both the diversity of samples (Pass Rate) and the speed of generation (Throughput).

| Method (N) | Dynamic Sampling Pass Rate | Generation Throughput (samples/s) |
| --- | --- | --- |
| ProRL (16) | 41% | 36.5 |
| BroRL (Ours, 512) | 62% | 72.4 |

samples are deemed useful for training, compared to only 41% for $N = 16$. This minimizes wasted computation and ensures each training step is based on more effective data.

Second, at the hardware level, our approach achieves a significantly higher generation through-put—nearly 100% faster (72.4 vs 36.5 samples/s). This improvement comes from addressing a common bottleneck in GPU computing: being memory-bound (Recasens et al., 2025). With small batches ($N = 16$), the generation process is often memory-bound; the GPU's compute cores idle while waiting to fetch data from memory. By generating a large number of samples ($N = 512$) at once, the operation becomes more compute-bound and also leads to a higher prefix cache hit rate (Zheng et al., 2024). This allows the GPU to leverage its parallel processing cores to their full poten-tial, increasing arithmetic intensity and leading to higher sustained computing utilization. Therefore, BroRL is not only a more effective RL training recipe but also utilizes the underlying hardware more efficiently.

## 5 RELATED WORK

**Reinforcement Learning for Reasoning** Reasoning models represent a specialized category of AI systems that engage in a long chain-of-thought to generate answers. This approach is central to models like DeepSeek-R1, which use RLVR as a core training methodology. The RLVR paradigm, which adapts RLHF techniques (Christiano et al., 2017; Ouyang et al., 2022), has popularized al-gorithms such as GRPO (Shao et al., 2024), RLOO (Ahmadian et al., 2024), REINFORCE++ (Hu et al., 2025a) and DAPO (Yu et al., 2025). However, RL training is notoriously sensitive to hyperpa-rameters, making stable, long-term optimization a significant challenge. While many open-source efforts exist, most focus on narrow domains or test-time compute scaling. Few have addressed the challenge of prolonged RL training or investigated the underlying training-time scaling laws, leaving a critical gap in understanding how to robustly enhance model reasoning.

**Scaling Axes in Reinforcement Learning** The scaling laws of the RL process itself are under-explored. Prior work has focused on the axis of the total number of training steps. For example, ProRL demonstrates that prolonged RL training can expand the reasoning boundaries of LLMs (Liu et al., 2025a). In contrast, we investigate a complementary axis: rollout size $N$, the number of rollouts ($N$) sampled per prompt in each update step. Our work, BroRL, is the first to formalize rollout size $N$ as a principled scaling dimension in RLVR. We provide a formal analysis proving that increasing $N$ dampens a negative "unsampled coupling" term in the policy update, ensuring a more reliable learning signal. This mechanism directly addresses the training instabilities that can limit RL's effectiveness for reasoning.

## 6 CONCLUSION

This work establishes rollout size $N$, not just longer steps, as a critical and efficient axis for scaling reinforcement learning in large language models. We demonstrated that the performance plateaus encountered by steps-scaling methods like ProRL are not fundamental limits but artifacts of an un-stable learning signal caused by insufficient exploration. Our theoretical analysis pinpointed the "unsampled coupling" term as the primary source of this instability and proved that increasing roll-out size $N$ systematically mitigates it. Empirically, our BroRL framework validated this theory by transforming a stagnated model into one capable of continuous learning, achieving state-of-the-art 1.5B model in complex reasoning tasks. Critically, these gains were achieved with superior computational efficiency, doubling hardware throughput by shifting the bottleneck from memory to compute in some cases, underscoring BroRL's practicality for real-world deployment.

## ETHICS STATEMENT

We acknowledge and adhere to the ICLR Code of Ethics. Our work introduces a more computationally efficient method for advancing AI reasoning. While this has the positive potential to democratize research and accelerate scientific discovery, we also recognize the associated risks. These include the dual-use of enhanced reasoning in sensitive domains like cybersecurity, the generation of sophisticated misinformation, and long-term socioeconomic impacts. Therefore, we advocate for the responsible development and deployment of our technology and stress that advancements in AI capabilities must be accompanied by parallel, robust efforts in safety, governance, and ethics research.

## REPRODUCIBILITY STATEMENT

We are committed to ensuring the reproducibility of our work. Our theoretical claims, including the derivation of the correct-mass change in Theorem 1 and the expected decay of unsampled mass in Lemma 2, are accompanied by detailed proofs, which are provided in Appendix D. Our experimental setup, detailed in Section 4, is built for reproducibility. We continue training on the same datasets (math, code, science, IFEval and reasoning gym) as the ProRLv2 baseline. Key training hyperparameters include a learning rate of 2e-6, weight decay of 0.01 with AdamW optimizer (Loshchilov & Hutter, 2019), and a KL coefficient of 0.0002 with a K2 loss type. We use 8 PPO mini-batches, with a *clip-high* of 0.28, *clip-low* of 0.2, and a truncated importance sampling ratio cap of 2. All sample generation was performed with a temperature of 1.0. The veRL environment was built on PyTorch 2.7 and vLLM 0.10.1. The core algorithmic components of our framework are described in Section 3.

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

## A  THE USE OF LARGE LANGUAGE MODELS

We utilized the Gemini 2.5 Pro to assist with English language polishing. This assistance was primarily applied to parts of the Abstract, Introduction, Methodology, and Experiments sections to improve grammar, clarity, and readability. All scientific content, ideas, and conclusions remain the original work of the authors.

## B  LIMITATIONS

A primary limitation of our current study is the scope of our investigation into the hyperparameter $N$, the rollout size. Our experiments focus on demonstrating the significant performance and efficiency gains achieved by moving from a small-rollout-size regime $N = 16$) to a large-rollout-size one ($N = 512$). While these results strongly support our central thesis, they do not provide a complete picture of the relationship between rollout size $N$ and model improvement.

A comprehensive analysis sweeping across a wider range of intermediate $N$ values (e.g., 64, 256, 1024) would be necessary to fully characterize this relationship. Such an analysis could reveal the precise shape of the performance curve, identify potential points of diminishing returns, and establish a more formal cost-benefit trade-off. Our simulation results (Figure 3) suggest that the gains are monotonic but concave, yet validating this trend on large-scale language models is a computationally demanding task that we leave for future work. A more granular understanding of this scaling behavior would provide invaluable practical guidance for researchers and practitioners aiming to select an optimal $N$ for their specific computational budget and performance targets.

## C  BROADER IMPACT

The development of more capable and efficient methods for training AI models, such as BroRL, has the potential for significant positive and negative societal impacts.

**Potential Positive Impacts.**  Our work demonstrates a path toward more computationally efficient scaling of reinforcement learning for LLMs. By improving sample efficiency and hardware utilization, BroRL could lower the barrier to entry for training highly capable reasoning models. This could democratize access to state-of-the-art AI, enabling academic institutions and smaller organizations to contribute to cutting-edge research. Furthermore, enhancing the mathematical, logical, and coding abilities of LLMs can accelerate scientific discovery, create more effective educational tools, and augment human expertise in complex technical domains.

**Potential Negative Impacts and Societal Risks.**  Like any advancement that increases the capabilities of AI systems, this work warrants a thoughtful consideration of potential risks. Enhanced reasoning and coding capabilities are powerful tools that could be applied in sensitive domains. For instance, the application of highly autonomous systems in areas such as cybersecurity requires careful oversight to prevent unintended consequences. Additionally, the ability to generate highly plausible and complex content at scale has implications for the information ecosystem that merit ongoing study. As with any powerful automation technology, the long-term economic and labor market impacts also warrant careful consideration by the broader community. It is crucial that the advancement of AI capabilities, spurred by research like ours, is accompanied by a parallel and robust effort in safety and ethics. We advocate for the responsible development and deployment of these models within a strong ethical framework.

## D  PROOF DETAILS

### D.1  THEOREM 1

**Notation.**  For clarity, we repeat key quantities: (i) $A, B, U$: sampled correct, sampled incorrect, and unsampled token sets. (ii) $Q_{\text{pos}}, Q_{\text{neg}}$: global correct/incorrect probability masses. (iii) $A_2, B_2, U_{\text{pos},2}, U_{\text{neg},2}$: second moments. (iv) . $S_R == R_c\,P_{\text{pos}} + R_w\,P_{\text{neg}}$ which represents the

net contribution of sampled tokens, balancing the rewards from correct and incorrect tokens. Define the reward $R_j$ for sampled correct, sampled incorrect and unsampled tokens as:

$$R_j = \begin{cases} R_c, & j \in A, \\ R_w, & j \in B, \\ 0, & j \in U. \end{cases}$$

**Logit update and Jacobian expansion.** We start from the TRPO-style (Schulman et al., 2015) linear surrogate

$$L_{\text{RLVR}}(\theta) = -\mathbb{E}_{x \sim \mathcal{D}}\Big[ \sum_y r(x,y)\,\pi_\theta(y \mid x) \Big] \approx -\frac{1}{N} \sum_{i \in A \cup B \cup U} R_i p_i,$$

where $R_i \in \{R_w, 0, R_c\}$. This linear surrogate furnishes a convenient Monte-Carlo estimate - sample average approximation when using a relative entropy - Kullback-Leibler regularizer. This estimator is unbiased, hence all derivation and integration operations carry through to be interchanged with the expectation sign (Asmussen & Glynn, 2007).

Denote $z_j$ as the logit for the $j$-th token. Then we differentiating w.r.t. $z_j$ using $\frac{\partial p_i}{\partial z_j} = p_i(\delta_{ij} - p_j)$ gives

$$\Delta z_j = \frac{\eta}{N} p_j(R_j - S_R), \quad S_R = \sum_{k \in A} p_k - \sum_{k \in B} p_k.$$

**First-order change in probabilities.** By first-order expansion,

$$\Delta p_i = \sum_{j=1}^{V} \frac{\partial p_i}{\partial z_j}\,\Delta z_j = p_i\Big(\Delta z_i - \sum_{j=1}^{V} p_j \Delta z_j\Big).$$

Summing over any index set $\mathcal{S}$,

$$\sum_{i \in \mathcal{S}} \Delta p_i = \sum_{i \in \mathcal{S}} p_i \Delta z_i - \Big(\sum_{i \in \mathcal{S}} p_i\Big)\Big(\sum_{j=1}^{V} p_j \Delta z_j\Big).$$

We will need

$$\sum_{j=1}^{V} p_j \Delta z_j = \frac{\eta}{N}\Big[(R_c - S_R)A_2 + (R_w - S_R)B_2 - S_R\,U_2\Big],$$

and, restricted to correct tokens,

$$\sum_{i \in \mathcal{P}} p_i \Delta z_i = \frac{\eta}{N}\Big[(R_c - S_R)A_2 - S_R\,U_{\text{pos},2}\Big].$$

**Total change of correct mass.** The total change of correct-token probability mass is

$$\Delta P_{\text{correct}} \equiv \sum_{i \in \mathcal{P}} \Delta p_i = \sum_{i \in \mathcal{P}} p_i \Delta z_i - Q_{\text{pos}} \sum_{j=1}^{V} p_j \Delta z_j.$$

Substituting the identities above and simplifying with $Q_{\text{pos}} = P_{\text{pos}} + P_{\text{pos,out}}$, $Q_{\text{neg}} = 1 - Q_{\text{pos}}$, and $U_2 = U_{\text{pos},2} + U_{\text{neg},2}$, we obtain the compact form

$$\boxed{\Delta P_{\text{correct}} = \frac{\eta}{N}\Big[(R_c - S_R)\,Q_{\text{neg}}\,A_2 \;+\; (S_R - R_w)\,Q_{\text{pos}}\,B_2 \;+\; S_R\big(Q_{\text{pos}}\,U_{\text{neg},2} - Q_{\text{neg}}\,U_{\text{pos},2}\big)\Big],}$$

$$(1)$$

with $S_R = R_c P_{\text{pos}} + R_w P_{\text{neg}}$.

## D.2  Lemma 2

We seek to obtain the scaling of the the unsampled second-moment with respect to $N$. For this, we work under the simple assumption of token drawn independently and identically distributed as Bernoulli random variables. This is a popular assumption (see e.g. Du et al. (2025)), which allows us to obtain a convenient analytical formula capturing the scaling we are interested in. This scaling is further corroborated by the extensive experimental results from Section 4.

Let $X \sim \text{Bin}(N, p)$ be the number of times a token is drawn in $N$ independent Bernoulli trials, each with success probability $p$. By the binomial distribution, the probability of never drawing the token is

$$\Pr[X = 0] = (1 - p)^N.$$

Equivalently, by independence across draws, the probability that the token is not selected in any of the $N$ trials is also $(1 - p)^N$. Define the indicator variable $I = \mathbf{1}\{X = 0\}$, which is 1 if the token is never sampled and 0 otherwise. The token's unsampled second-moment contribution is then the random variable

$$S = p^2 I.$$

Taking expectations, we obtain

$$\mathbb{E}[S] = p^2 \, \mathbb{E}[I] = p^2 \, \Pr[I = 1] = p^2 \, \Pr[X = 0] = p^2 (1 - p)^N.$$

## E  Empirical Evaluation

To rigorously test whether rollout size $N$ scaling breaks the training–depth plateau observed at 3,000 RL steps in the baseline (Liu et al., 2025a), we compare *ProRL* (small rollout size $N = 16$ and longer steps) against *BroRL* (large rollout size $N = 512$) under an identical evaluation protocol across three task families: math competitions (AIME/AMC, MATH, Minerva, OlympiadBench (of America, 2024; 2025; Hendrycks et al., 2021b; Lewkowycz et al., 2022; He et al., 2024)), code generation (APPS, CodeContests/Codeforces, TACO (Hendrycks et al., 2021a; Li et al., 2022; 2023)), and multi-domain reasoning (Reasoning Gym (Stojanovski et al., 2025)). Importantly, the table columns capture *training* controls: $N$ is the number of samples per prompt, $B$ is the number of prompts per RL step, and Steps is the count of continued RL steps. For details on sample generation and GPU compute consumption, please refer to Table 1. For evaluation, we report **pass@1** with a 32k context length, averaged over 16 independent samples per instance to ensure stable estimates, using nucleus sampling (top_p=0.95) with a temperature of 0.6.

The experimental results presented in the tables 3,4,5 unequivocally support the superiority of rollout size $N$ scaling. In the critical domain of mathematical reasoning, the *ProRL* approach confirms the performance plateau; after an initial small gain (from a 61.69 baseline to 62.08), its average score slightly degrades to 62.02. In stark contrast, *BroRL* not only avoids this degradation but also consistently improves, reaching a superior score of 62.85 after 134 steps. This advantage is even more pronounced in other domains. For code generation, BroRL's score jump (+1.48 points) far exceeds the marginal gains from ProRL (+0.74 points). Similarly, on the Reasoning Gym benchmark, BroRL achieves a substantial improvement of over 1.5 points, while ProRL provides almost no meaningful gain.

In conclusion, across all three demanding domains, widening the generation search space per RL step proves to be a significantly more effective and efficient strategy than merely continuing training with a narrow search. Crucially, as detailed in Table 1, BroRL achieves these superior results with a comparable number of total generated samples while consuming fewer wall-clock GPU hours. The BroRL method successfully overcomes the performance limitations observed in the baseline, leading to stronger and more stable reasoning capabilities. This highlights that for complex problem-solving, the diversity of experience in each training step is more crucial than the sheer length of the training process.

Table 3: Math scores.

| Method | N | B | Steps | AIME24 | AIME25 | AMC | Math | Minerva | Olympiad Bench | Math Avg. |
|--------|---|---|-------|--------|--------|-----|------|---------|----------------|-----------|
| Baseline | 16 | 512 | 3000 | 49.58 | 36.04 | 82.53 | 92.49 | 49.03 | 60.44 | 61.69 |
| ProRL | 16 | 512 | +225 | 54.58 | 36.25 | 80.95 | 91.93 | 48.25 | 60.52 | 62.08 |
| ProRL | 16 | 512 | +535 | 54.38 | 35.83 | 80.42 | 92.15 | 48.55 | 60.77 | 62.02 |
| BroRL | 512 | 128 | +107 | 56.10 | 35.30 | 81.76 | 92.18 | 48.92 | 61.41 | 62.62 |
| BroRL | 512 | 128 | +134 | 57.71 | 35.63 | 80.12 | 92.06 | 49.72 | 61.87 | 62.85 |
| BroRL | 512 | 128 | +191 | 57.50 | 36.88 | 81.02 | 92.14 | 49.08 | 61.54 | **63.03** |

Table 4: Code generation scores.

| Method | N | B | Steps | apps | codecontests | codeforces | taco | Code Avg. |
|--------|---|---|-------|------|--------------|------------|------|-----------|
| Baseline | 16 | 512 | 3000 | 58.52 | 54.99 | 58.64 | 35.87 | 52.00 |
| ProRL | 16 | 512 | +225 | 58.83 | 54.58 | 59.27 | 36.36 | 52.26 |
| ProRL | 16 | 512 | +535 | 59.67 | 55.09 | 59.13 | 37.06 | 52.74 |
| BroRL | 512 | 128 | +107 | 60.28 | 55.84 | 59.80 | 37.31 | 53.31 |
| BroRL | 512 | 128 | +134 | 60.19 | 56.52 | 60.04 | 37.15 | 53.48 |
| BroRL | 512 | 128 | +191 | 61.59 | 56.62 | 60.86 | 37.74 | **54.20** |

Table 5: Reasoning Gym scores.

| Method | N | B | Steps | algebra | algorithmic | arc | arithmetic | code | cognition | games | geometry | graphs | induction | logic | Avg. |
|--------|---|---|-------|---------|-------------|-----|------------|------|-----------|-------|----------|--------|-----------|-------|------|
| Baseline | 16 | 512 | 3000 | 97.19 | 55.32 | 4.98 | 85.74 | 48.20 | 45.91 | 25.68 | 91.62 | 70.25 | 80.25 | 82.25 | 61.29 |
| ProRL | 16 | 512 | +225 | 97.01 | 58.22 | 5.33 | 85.74 | 47.96 | 46.01 | 25.55 | 91.59 | 69.83 | 80.31 | 85.26 | 62.10 |
| ProRL | 16 | 512 | +535 | 97.46 | 55.56 | 4.79 | 85.70 | 48.43 | 46.33 | 25.71 | 92.56 | 70.40 | 80.31 | 85.29 | 61.45 |
| BroRL | 512 | 128 | +107 | 97.55 | 59.11 | 5.10 | 85.97 | 49.22 | 44.05 | 25.99 | 92.16 | 71.51 | 80.40 | 85.41 | 62.71 |
| BroRL | 512 | 128 | +134 | 97.70 | 59.28 | 5.31 | 85.95 | 49.30 | 44.53 | 25.88 | 92.88 | 72.01 | 80.38 | 85.29 | 62.82 |
| BroRL | 512 | 128 | +191 | 97.59 | 59.65 | 6.27 | 86.17 | 49.45 | 45.51 | 25.77 | 93.00 | 72.03 | 80.94 | 85.56 | **63.09** |

