# OpenReview forum: "BroRL: Scaling Reinforcement Learning via Broadened Exploration"
_ICLR.cc/2026/Conference — Submitted to ICLR 2026_

### Official Review · Reviewer_PnHZ · 2025-10-28

**Soundness:** 3
**Presentation:** 2
**Contribution:** 2
**Rating:** 4
**Confidence:** 4

**Summary:**

This paper introduces BroRL, a new paradigm for scaling RLVR. Unlike ProRL, which improves model performance by increasing training steps but eventually plateaus, BroRL enhances performance by increasing the number of rollouts per example, i.e., broadening exploration to achieve continuous gains even after ProRL saturation. The method is grounded in a probability mass balance analysis, showing that as rollout numbers grow, the model consistently expands the probability mass of correct tokens while minimizing the influence of unsampled terms. Empirical results confirm that BroRL revives models that have stagnated after thousands of ProRL steps, achieving SOTA results for 1.5B-parameter models across diverse benchmarks. Moreover, under the same training time, BroRL proves to be more data- and compute-efficient, nearly doubling throughput by shifting generation from memory-bound to compute-bound execution, demonstrating strong practicality for real-world deployment.

**Strengths:**

1. Novel scaling paradigm: this work introduces a new way to scale RL via more rollouts per example, that complements existing methods like ProRL, offering an alternative to merely increasing training steps.


2. Theoretical grounding: the paper provides a clear analytical framework by mass balance equation analysis, explaining why increased rollouts improve the probability mass of correct tokens.


3. Empirical validation: theoretical findings in this paper are supported by simulations and experiments showing consistent performance gains, even after ProRL saturation.


4. Efficiency gains: the experimental results demonstrate higher data and compute efficiency; by shifting computation from memory-bound to compute-bound, BroRL nearly doubles throughput under the same hardware conditions.

**Weaknesses:**

1. Limited theoretical scope: the main analysis relies on a one-step RL assumption, which may not fully generalize to multi-step or real-world RL settings. It would be helpful if the authors can provide a discussion about the extension to multi-step RL settings.


2. Empirical range: validation is mainly conducted on mid-sized models, i.e., 1.5B parameters. Scalability of BroRL to larger models or a broader range of tasks remains to be demonstrated, particularly since efficiency is a key concern for this method.


3. Naive approach: the core idea of BroRL, i.e., simply increasing the number of sampled rollouts per example, is relatively straightforward. While effective, it may be considered a naive method, as improved performance could arise largely from brute-force sampling rather than a more principled algorithmic innovation.


4. Limited exploration techniques: many alternative strategies for encouraging sampling diversity or more efficient exploration are not considered or compared. It is unclear whether similar performance gains could be achieved without resorting to large-scale rollouts, for example, by using smarter exploration or diversity-promoting methods.


Minor point: Figure 2 is not immediately clear, even though its meaning becomes understandable from the later context. It would be helpful to clarify the figure or make it more self-explanatory.

**Questions:**

Q1. In Figure 1, does the x-axis represent training time only, or does it also include sample generation time? If it represents training time only, its significance may be limited, since the number of training samples for ProRL is relatively small.

---

> ### Author Response · Authors · 2025-11-19
>
> ### weaknesses 1
>
> We thank the reviewer and agree that our theoretical analysis relies on a one-step RL assumption. This simplification was chosen to clearly isolate the dynamics of sampled versus unsampled token probabilities and led to the identification of the “unsampled coupling” term as a key source of instability. The theory shows that increasing rollout size \(N\) systematically suppresses this term.
>
> To test the robustness of this insight, we first ran token-level simulations (Figure 3). These confirmed that larger \(N\) yields more stable updates, faster growth of correct probability mass, and elimination of “knowledge shrinkage” observed at small \(N\), providing a bridge from the one-step theory to more realistic dynamics. Our large-scale experiments then validate the principle in a full multi-step PPO setting: ProRL with small \(N\) (16) receives a sparse, high-variance signal from a narrow slice of trajectory space, while BroRL with large \(N\) (512) obtains a richer, lower-variance signal. BroRL’s ability to break through the 3K-step saturation of ProRL provides strong evidence that the core mechanism remains relevant in practical multi-step RL.
>
> ### weaknesses 2
>
> We reran experiments on Qwen‑3 4B Instruct and found that it quickly saturates on the ProRLv2 setup. Extending ProRLv2 training caused a clear Math score drop (71.65 → 69.58), indicating instability after saturation. In contrast, BroRL with 30 additional steps maintained roughly monotonic performance and avoided regressions, showing that our method scales to larger models while preserving the stable training behavior predicted by our analysis.
>
> | Method                   | RL Steps | Math Score | Code Score | Reasoning Gym Score |
> |--------------------------|----------|------------|------------|---------------------|
> | Qwen3-4B-Instruct-2507   | 0        | 68.22      | 51.77      | 28.97               |
> | ProRLv2 (N=16, B=512)    | 333      | 71.65      | 64.79      | 77.95               |
> | ProRLv2 (N=16, B=512)    | 396      | 69.58      | 66.15      | 78.15               |
> | BroRL (N=256, B=128)     | 333+30   | 71.64      | 65.54      | 78.13               |
>
> We will add these results to the final paper.
>
> ### weaknesses 3
>
> We view BroRL’s simplicity as a strength. The main contribution is the theoretical analysis showing why increasing rollout size \(N\) is effective and not a brute-force trick. Our mass-balance analysis is, to our knowledge, the first to formally diagnose RLVR performance dynamics and attribute observed “artifacts” to an unstable, noisy unsampled coupling term. The theory proves that scaling \(N\) is the direct mechanism to suppress this term and stabilize the learning signal by improving the balance between correct (\(Q_{\text{pos}}\)) and incorrect (\(Q_{\text{neg}}\)) token mass. This provides a simple, theoretically grounded prescription—larger \(N\)—for tuning RLVR systems.
>
> ### weaknesses 4
>
> BroRL is intended as a complementary scaling axis rather than a replacement for “smarter” exploration methods. Our contribution is to identify a statistical pathology in small-batch RL updates: the unsampled coupling term that destabilizes gradients and causes “knowledge shrinkage,” where correct solutions are forgotten. Theorem 1 and simulations (Figure 3) show that increasing rollout size \(N\) directly mitigates this term and preserves correct-token probabilities.
>
> Empirically, BroRL is both data- and compute-efficient. Scaling to large \(N\) (e.g., 512) moves the system from memory-bound to compute-bound and nearly doubles throughput (72.4 vs. 36.5 samples/s) on identical hardware. A larger, more representative batch also raises the Dynamic Sampling Pass Rate from 41% (\(N=16\)) to 62% (\(N=512\)), reducing wasted computation on trivial trajectories. Thus, large-\(N\) rollouts are an efficient scaling strategy. Exploration/diversity methods mainly address how to find good trajectories, whereas BroRL focuses on how to stably learn from the sampled batch; these directions are orthogonal and potentially synergistic.
>
> ### Minor point
>
> Figure 2 aims to decompose the one-step change in correct-token probability into “Sampled” (always non-negative) and “Unsampled” (conditional) components, and to illustrate how larger \(N\) reduces the unstable unsampled term. We will revise the caption to explain these components and their interaction more clearly so that the figure is self-contained.
>
> ### Questions 1
>
> In Figure 1, the x-axis is total wall-clock time, including both rollout (sample generation) and training (gradient update). Each RL step inherently consists of both phases, so the x-axis reflects total compute. The figure’s main goal is to show that BroRL overcomes the performance plateau where ProRL saturates and even degrades; detailed algorithmic and hardware efficiency comparisons are provided separately in Table 2.

---

> > ### Comment · Reviewer_PnHZ · 2025-11-26
> > **Reviewer response**
> >
> > Thank you for your response. I acknowledge and appreciate the contribution of your theoretical analysis. However, I still find the overall contribution somewhat incremental. As mentioned previously, simply increasing the sampling seems like a rather naive idea, and both the theoretical and experimental results primarily confirm an expected outcome.

---

### Official Review · Reviewer_EGGu · 2025-10-29

**Soundness:** 2
**Presentation:** 3
**Contribution:** 2
**Rating:** 2
**Confidence:** 3

**Summary:**

This paper proposes BroRL, a reinforcement learning framework that scales the number of rollouts per prompt to improve training performance. The authors develop a mass transfer model to explain how increasing the rollout size N reduces the coupling of unsampled probability mass, thereby increasing the positive probability shift $\Delta Q_{pos}$. Experiments conducted on a token-level simulator and following the ProRL training setup show a small but statistically significant improvement in test scores. The paper also claims improved hardware utilization and generation throughput compared to standard RL methods.

**Strengths:**

- The mass transfer modeling is insightful. It provides a way to analyze the probability change in RL.
- Under the same GPU computing budget, BroRL achieves better performance than ProRL.

**Weaknesses:**

- Limited novelty. The core idea is simply increasing the rollout size from 16 to 512, which is incremental. The paper lacks a clear rationale for choosing 512 and does not systematically study how varying N affects RL dynamics. Discussion of moderate (e.g., 64) or extreme (e.g., 2048) rollout sizes is missing.

- Efficiency concerns. Prior work [1][2] shows that in later training stages, easy problems dominate. In BroRL, generating 512 rollouts for these trivial cases may waste substantial computation. While the paper reports higher generation throughput, it does not clearly link this to actual learning efficiency or performance improvement.

[1] Act Only When It Pays: Efficient Reinforcement Learning for LLM Reasoning via Selective Rollouts

[2] POLARIS: A POst-training recipe for scaling reinforcement Learning on Advanced ReasonIng modelS

**Questions:**

1. In the mass transfer analysis, larger rollout sizes seem consistently beneficial. Do the authors agree that "larger is always better"? Have you experimented with rollout sizes greater than 512?

2. During training, with 512 rollouts per question, what is the distribution of correct versus incorrect samples? Does the "easy-problem dominance" phenomenon mentioned in [1][2] also appear in BroRL?

---

> ### Author Response · Authors · 2025-11-19
>
> ### weaknesses 1
>
> We thank the reviewer and agree that BroRL’s simplicity is a strength. The key contribution is the underlying analysis that explains why increasing rollout size \(N\) is effective and compute-efficient, rather than a brute-force trick. Scaling \(N\) shifts the hardware bottleneck from memory-bound to compute-bound and nearly doubles generation throughput in our setup, making BroRL practical for deployment.
>
> Our mass-balance analysis is, to our knowledge, the first to formally characterize performance dynamics in RLVR. It shows that saturation arises from an unstable learning signal driven by a noisy “unsampled coupling” term. The theory proves that increasing \(N\) directly suppresses this term, stabilizing the learning signal and improving the balance between correct (\(Q_{\text{pos}}\)) and incorrect (\(Q_{\text{neg}}\)) token mass. This gives a principled explanation for why a simple change in \(N\) substantially improves training.
>
> We conducted an ablation with \(N=256\), showing a clear trend \(N=512 > N=256 > N=16\). Due to a 4-hour per-RL-step constraint, \(N=512\) is the largest value feasible in our setup; training the 1.5B models already consumed over 41,216 H100 GPU hours.
>
> | Method  | N   | RL Steps   | Total Time (h) | Math Score | Code Score | Reasoning Gym Score |
> |---------|-----|------------|----------------|------------|------------|---------------------|
> | Baseline| 16  | 3000       | -              | 61.69      | 52.00      | 61.29               |
> | ProRL   | 16  | 3000+535   | 133.8          | 62.02      | 52.74      | 61.45               |
> | BroRL   | 256 | 3000+202   | +118.2         | 62.23      | 53.43      | 61.78               |
> | BroRL   | 512 | 3000+134   | +122.8         | 62.85      | 53.48      | 62.82               |
>
> BroRL also achieves higher accuracy with fewer output tokens on Math and Code, indicating better score-per-token efficiency and more concise reasoning:
>
> | Task | ProRL Score | BroRL Score | Score Diff | ProRL Tokens | BroRL Tokens | Token Diff |
> |------|-------------|-------------|------------|--------------|--------------|------------|
> | Math | 62.02       | 63.66       | +1.64      | 16,506       | 15,760       | -745       |
> | Code | 52.74       | 56.64       | +3.90      | 26,808       | 26,090       | -717       |
>
> We will integrate these results into the final version of the paper.
>
> ### weaknesses 2
>
> We agree that later-stage training can include easier tasks and that naive sampling might seem wasteful. BroRL addresses this while remaining simple to deploy.
>
> First, training data can be pre-filtered to remove trivial items before RL. Second, dynamically reallocating sampling budgets between easy and hard tasks would require a multi-stage procedure (e.g., a small-\(N\) probe followed by adaptive resampling). This reduces effective batch size and harms GPU utilization, so the theoretical savings may be limited on modern hardware.
>
> Our goal is to keep BroRL easy to integrate into existing RL frameworks without complex code changes. Importantly, large-\(N\) rollouts in BroRL are not computationally wasteful: they are both data- and compute-efficient. At the hardware level, scaling \(N\) (e.g., to 512) moves the system from memory-bound to compute-bound, nearly doubling throughput (72.4 vs. 36.5 samples/s on identical hardware). At the algorithmic level, the larger, more representative batch increases the Dynamic Sampling Pass Rate from 41% (\(N=16\)) to 62% (\(N=512\)), reducing time spent on uninformative trajectories. Thus, large-\(N\) rollouts constitute a more efficient scaling strategy, not a brute-force one.
>
> We view “smarter exploration” and diversity-promoting methods as complementary. Those techniques focus on finding better trajectories, while BroRL focuses on stabilizing learning from the trajectories given. Combining these axes is a natural direction for future work, as also noted in our limitations.
>
> ### Questions 1
>
> Theoretical analysis and simulations indicate that larger rollout sizes consistently stabilize the learning signal: as \(N\) increases, the negative unsampled coupling term shrinks, making the policy update more reliable. We do not claim linear gains and explicitly acknowledge potential diminishing returns.
>
> In token-level simulations, we explored very large \(N\) (up to 51,200) and observed that sufficiently large \(N\) can eliminate “knowledge shrinkage.” For full-scale language models, we focused on the practically important regime from \(N=16\) to \(N=512\), constrained by a 4-hour per-RL-step limit; \(N=512\) is the largest value feasible under this constraint. Training the 1.5B models required more than 41,216 H100 GPU hours.
>
> ### Questions 2
>
> In the later stages of training, easier samples did increase but did not dominate the dataset. For the 1.5B model, the final overall rollout sample accuracy was approximately 0.65.

---

### Official Review · Reviewer_gy8t · 2025-10-31

**Soundness:** 2
**Presentation:** 3
**Contribution:** 2
**Rating:** 6
**Confidence:** 2

**Summary:**

This paper investigates a novel dimension for scaling Reinforcement Learning with Verifiable Rewards in large language models. The authors posit that increasing the number of rollouts per prompt ($N$) to a large number (e.g., 512), rather than solely scaling the number of training steps, can lead to more stable and performant policy optimization. They provide a theoretical analysis based on a "mass balance equation" to show that a large N suppresses a potentially negative "unsampled coupling" term in the policy update, guaranteeing an increase in the probability mass of correct tokens. Empirically, they demonstrate that applying BroRL to a saturated ProRL model yields continued performance improvements on reasoning benchmarks and offers computational efficiency gains by shifting the GPU bottleneck from memory to compute.

**Strengths:**

- The idea of scaling the "width" of exploration (rollouts per prompt) instead of just the "depth" (training steps) represents an under-explored direction in RL scaling literature. It directly addresses the common problem of performance plateauing in prolonged RL training.

- The paper provides a theoretical foundation for its claims. The mass balance equation and the decomposition of $\Delta Q_{pos}$ offer a clear, mechanistic explanation for why increasing $N$ should lead to more stable learning by mitigating the variance from unsampled tokens.

- The experiments are generally well-designed to support the core thesis. The token-level simulation cleanly validates the theoretical dynamics, and the successful application to a large-scale model that had already plateaued is a strong, practical demonstration of BroRL's potential.

**Weaknesses:**

- Incomplete Ablation on $N$: The paper jumps from $N=16$ (ProRL) to $N=512$ (BroRL). A more granular ablation study with intermediate N values (e.g., 64, 128, 256) is missing. This would be crucial for understanding the scaling law's shape, identifying potential diminishing returns, and providing practical guidance for choosing N. The authors acknowledge this in Appendix B, but it weakens the current analysis.

- All large-scale empirical results are based on a 1.5B parameter model. This severely limits the generalizability of the findings. It is well-established in LLM literature that scaling laws and algorithmic behaviors can differ significantly between small models and larger-scale models. The claim that BroRL is a general scaling law for RLVR is not fully substantiated without validation on larger model sizes.

**Questions:**

- The theoretical analysis is performed with a TRPO-style linear surrogate. To what extent do you believe the insights from Theorem 1 hold for the actual clipped PPO/GRPO objective with a KL penalty?

- Why was N=512 chosen for BroRL? Was this based on preliminary scaling experiments, hardware constraints, or intuition? Given the lack of a sweep over N, it is unclear if this is an optimal or merely a sufficient value.

- How BroRL works on other RL algorithms (e.g. DAPO)?. The related work section acknowledges ProRL's step-scaling but does not sufficiently situate BroRL against other RL algorithm families.

---

> ### Author Response · Authors · 2025-11-19
>
> ### weaknesses 2
>
> We thank the reviewer for this point and agree that validation on larger models is essential.
>
> To address this, we ran additional experiments on a 4B-parameter model (Qwen3-4B-Instruct). On this larger model, the baseline ProRLv2 recipe quickly saturated and then became unstable: while it initially improved performance (step 333), continued training degraded the Math score from 71.65 to 69.58. In contrast, BroRL maintained a stable trajectory and avoided significant drops, confirming better training stability at scale.
>
> | Method                     | RL Steps | Math Score | Code Score | Reasoning Gym Score |
> |----------------------------|----------|------------|------------|---------------------|
> | Qwen3-4B-Instruct-2507     | 0        | 68.22      | 51.77      | 28.97               |
> | ProRLv2 (N=16, B=512)      | 333      | 71.65      | 64.79      | 77.95               |
> | ProRLv2 (N=16, B=512)      | 396      | 69.58      | 66.15      | 78.15               |
> | BroRL (N=256, B=128)       | 333+30   | 71.64      | 65.54      | 78.13               |
>
> These 4B results show that instability of standard RL training can worsen with scale, and that BroRL mitigates this effect, supporting our claim that BroRL is a more scalable solution. We will add these findings to the main paper.
>
> ### Questions 1
>
> Theorem 1 focuses on the direction of the policy update, which is orthogonal to the stability mechanisms of PPO/GRPO. PPO-style clipping and KL penalties constrain the magnitude of the policy change, ensuring each update step is small. Our analysis shows that even with a small step, the update direction can still be harmful (reducing correct-token mass) due to the unsampled coupling term.
>
> The analysis (Appendix C.1) is a first-order expansion, valid in the limit of an infinitesimally small update. PPO/GRPO’s clipping and KL penalties are designed to approximate this “tiny step” regime. In this limit, clipping is effectively inactive, allowing us to isolate the sign of the update. In summary, PPO/GRPO keeps the step small, while our theory and BroRL (via scaling \(N\)) ensure that this small step points in the correct direction by mitigating the influence of unsampled tokens.
>
> ### Questions 2 / weaknesses 1
>
> We thank the reviewer for suggesting a more granular analysis of \(N\). To address this, we added an ablation with \(N=256\), providing an intermediate point between \(N=16\) (ProRL) and \(N=512\) (BroRL). The results show a clear monotonic trend across all three benchmarks:
>
> | Method  | N   | RL Steps   | Total Time (h) | Math Score | Code Score | Reasoning Gym Score |
> |---------|-----|------------|----------------|------------|------------|---------------------|
> | Baseline| 16  | 3000       | -              | 61.69      | 52.00      | 61.29               |
> | ProRL   | 16  | 3000+535   | 133.8          | 62.02      | 52.74      | 61.45               |
> | BroRL   | 256 | 3000+202   | +118.2         | 62.23      | 53.43      | 61.78               |
> | BroRL   | 512 | 3000+134   | +122.8         | 62.85      | 53.48      | 62.82               |
>
> The ordering BroRL \(N=512 >\) BroRL \(N=256 >\) ProRL \(N=16\) supports our scaling claim. A denser sweep would be ideal but is prohibitive: the 1.5B experiments already consumed over 41,216 H100 GPU hours, and a strict 4-hour per-step limit makes \(N>512\) infeasible. Within these constraints, \(N=256\) still provides strong evidence for our conclusions, and we will surface this table more prominently.
>
> We also observe that BroRL attains higher accuracy with fewer output tokens on Math and Code, indicating better score-per-token efficiency and less redundant reasoning:
>
> | Task | ProRL Score | BroRL Score | Score Diff | ProRL Tokens | BroRL Tokens | Token Diff |
> |------|-------------|-------------|------------|--------------|--------------|------------|
> | Math | 62.02       | 63.66       | +1.64      | 16,506       | 15,760       | -745       |
> | Code | 52.74       | 56.64       | +3.90      | 26,808       | 26,090       | -717       |
>
> ### Questions 3
>
> We designed BroRL to be compatible with other RL algorithms, and our current setup already reflects this. The ProRLv2 recipe we build on incorporates key techniques from DAPO, including dynamic sampling and the “clip higher” method.
>
> BroRL’s contribution is to increase rollout size \(N\), improving the sampling process in a way that is orthogonal to these policy optimization techniques. As the experiments show, BroRL can be combined effectively with existing RL algorithms such as those in ProRLv2/DAPO.

---

### Official Review · Reviewer_iznC · 2025-11-02

**Soundness:** 3
**Presentation:** 2
**Contribution:** 2
**Rating:** 4
**Confidence:** 3

**Summary:**

BroRL reframes RLVR scaling by turning up rollouts per prompt (N) rather than training steps. A mass-balance analysis decomposes the change in correct-token probability into a nonnegative sampled term plus an “unsampled coupling” term whose influence fades as N grows, so larger N drives net expansion of correct mass.  Simulations with a TRPO-style surrogate corroborate that sufficiently large N eliminates knowledge shrinkage and monotonically increases correct-token mass.  Empirically, continuing a 1.5B model that plateaus under ProRL after ~3k steps, BroRL sustains improvements across math/code/multidomain benchmarks while ProRL stalls or degrades.  Practically, big-N rollouts raise dynamic-sampling pass rate (62% vs 41%) and nearly double generation throughput (72.4 vs 36.5 samples/s) by shifting generation from memory-bound to compute-bound, improving algorithmic and hardware efficiency.  Overall, the paper argues that exploration breadth—not step count—is the critical, implementation-friendly axis for stable, scalable RLVR, turning saturated training into continued gains with better compute utilization.

**Strengths:**

1. The proposed method is simple and shows effective improvement across diverse benchmarks over the baseline method ProRL.
2. This work provides a clear mass-balance decomposition and shows sampled terms are always non-negative and the unsampled coupling term shrinks with larger rollout size (N).
3. token-level TRPO-style experiments verify that big-N stabilizes updates, accelerates correct-mass growth, and eliminates knowledge-shrinkage.

**Weaknesses:**

1. The idea of "scaling the number of sampled responses" is not novel and has been proposed and studied in previous works [1,2], demonstrating its effectiveness. These works also prove that scaling the number of sampled responses can benefit the scaling trends of RL training.

2. The detailed training settings are not provided, including the base model, the starting point, and the training dataset. The lack of information hinders the reproduction and evaluation of this work.

3. The improvement is relatively marginal (~1% for most benchmarks) under similar GPU training hours. Though increasing the number of rollouts can boost further performance improvement in RL training, the performance gain is too marginal, which makes the contribution of this work less significant. The authors may further study how to achieve more significant scaling gains or demonstrate the effectiveness on larger models.

[1] Hou et al, T1: Advancing Language Model Reasoning through Reinforcement Learning and Inference Scaling.

[2] Shao et al, DeepSeekMath: Pushing the Limits of Mathematical
Reasoning in Open Language Models.

**Questions:**

See weakness.

---

> ### Author Response · Authors · 2025-11-19
>
> ### weaknesses 1
>
> Thank you for raising this point. Prior work such as T1 and DeepSeekMath has shown that increasing sampled responses can improve performance. Our work builds on this line but differs in theory, scope, and empirical focus.
>
> First, prior work mainly reports empirical gains and treats sampling as a heuristic, typically at inference time or within a fixed RL setup. In contrast, we provide a formal analysis of rollout-size scaling in RLVR. We identify a previously unreported unsampled coupling term that can induce instability or reduce correctness, and prove that increasing rollout size \(N\) suppresses this term, yielding non-negative correct-mass change in the limit. This mechanism has not, to our knowledge, been analyzed before.
>
> Second, our empirical results differ qualitatively. While T1 and DeepSeekMath show that sampling improves performance, our experiments show that rollout-size scaling breaks through the plateau of prolonged RL training, achieving continued gains where step-scaling stagnates. This behavior is predicted by our theory and confirmed in both simulations and large-scale RLVR training.
>
> Third, earlier works consider sampling for inference scaling or reward-model robustness, whereas we treat rollout size \(N\) as a core scaling axis of the RL training procedure itself, distinct from training steps. Large-\(N\) rollouts reshape optimization dynamics, reduce variance, suppress instability, and at scale shift GPU workload from memory-bound to compute-bound, nearly doubling throughput. These findings extend beyond [1, 2].
>
> In summary, prior work observes that more samples can help empirically; our contribution is to formalize and explain this as a principled scaling law for RLVR, with new theoretical and practical insights complementary to [1, 2]. We will add these works to the related work section.
>
> ### weaknesses 2
>
> Thank you for highlighting the need for clearer training settings and dataset links. We will explicitly provide the following details.
>
> #### Base model
>
> - Nemotron-Research-Reasoning-Qwen-1.5, initialized from:
>   https://huggingface.co/nvidia/Nemotron-Research-Reasoning-Qwen-1.5B
>
> #### Training datasets (~136k samples, ProRLv2 setup)
>
> - Math reasoning: https://huggingface.co/datasets/agentica-org/DeepScaleR-Preview-Dataset
> - Code: https://huggingface.co/datasets/PRIME-RL/Eurus-2-RL-Data
> - STEM: https://huggingface.co/datasets/EricLu/SCP-116K
> - Logical puzzles: https://github.com/open-thought/reasoning-gym
> - Instructions: https://huggingface.co/datasets/nvidia/Llama-Nemotron-Post-Training-Dataset/viewer/RL
>
> #### Hyperparameters for BroRL
>
> | Hyperparameter           | Value                          |
> |--------------------------|--------------------------------|
> | Reference policy reset   | About every 100 steps          |
> | KL regularization        | K2/MSE; coefficient \(2\times10^{-4}\) |
> | Prompt size per step     | 128                            |
> | Samples per prompt \(N\) | 512                            |
> | Learning rate            | \(2\times10^{-6}\)             |
> | Context length           | 16,384 tokens                  |
>
> ### weaknesses 3
>
> Because our experiments start from a strong 3,000-step baseline that had already plateaued, headroom for easy gains is limited. We therefore extended BroRL training beyond this point.
>
> The new results show that BroRL unlocks substantially larger gains from the saturated 3,000-step model than ProRL achieved from 2,000 to 3,000 steps, supporting the claim of robust, continued improvement where step-scaling shows diminishing or negative returns. The Code Score reaches 56.64, a +4.64 absolute gain over the 3,000-step baseline (nearly 9% relative):
>
> | Method | N   | RL Steps  | Total Time (h) | Math Score | Code Score | Reasoning Gym Score |
> |--------|-----|-----------|----------------|------------|------------|---------------------|
> | ProRL  | 16  | 2000      | -              | 60.14      | 51.43      | 59.06               |
> | ProRL  | 16  | 3000      | -              | 61.69      | 52.00      | 61.29               |
> | BroRL  | 512 | 3000+419  | +393.9         | 63.66      | 56.64      | 63.40               |
>
> For larger models, we ran Qwen-3 4B Instruct. Under the ProRL small-\(N\) recipe, further training with \(N=16\) reduced the Math score from 71.65 to 69.58, indicating over-saturation. In contrast, applying BroRL with \(N=256\) after saturation maintained roughly monotonic improvement and avoided notable regressions. This suggests that BroRL’s principles transfer to larger models and provide more stable, reliable training where step-scaling begins to fail.

---

### Meta-Review · Area_Chair_ZyH6 · 2026-01-05

**Summary:**

This paper proposes BroRL, arguing that scaling RLVR by increasing rollouts per prompt (large-N exploration) can continue improving performance after step-scaling (e.g., ProRL) saturates.
The work introduces a mass-balance analysis that decomposes correct-token probability change into a nonnegative sampled term and an “unsampled coupling” term that can be harmful but diminishes with larger N, motivating BroRL as a stability-oriented scaling axis. The empirical results on a 1.5B LLMs show continued gains where small-N training stalls or regresses, and the rebuttal adds intermediate-N evidence (N=256) as well as a 4B validation demonstrating improved stability vs continued ProRL training.
Reviewer opinions remain mixed. Positive feedback (gy8t) highlights the clarity of the theoretical decomposition, practical effectiveness on a saturated regime, and improved compute/utilization characteristics. Skeptical feedback (EGGu, PnHZ, iznC) centers on novelty, since increasing sampling has appeared in prior work, and on the limited characterization of the N-scaling curve (only a few N values) and learning-efficiency concerns for large rollouts in late-stage training.

**Reviewer Concerns:**

1. The method proposed in the paper lacks novelty (iznC, EGGu, PnHZ).

2. The systematic scaling law with respect to N is still insufficient, with only three data points (N=16/256/512) (EGGu, gy8t).

3. Increasing N introduces additional overhead (EGGu).

**Reviewer Scores:**

Reviewer EGGu and PnHZ are unlikely to change their scores, as the core concerns—novelty and efficiency—were not resolved.
Reviewer gy8t will likely maintain the score or slightly increase it, since the authors added intermediate N results and larger-model experiments during the rebuttal.
Reviewer iznC may slightly increase the score, as the authors committed to open-sourcing and added code and 4B-scale experiments during the rebuttal.

---

### Decision · Program_Chairs · 2026-01-26

Reject